

# Climate change favours connectivity between virus-bearing pest and rice cultivations in sub-Saharan Africa, depressing local economies

Mattia Iannella, Walter De Simone, Paola D'Alessandro and Maurizio Biondi

Department of Life, Health & Environmental Sciences, University of L'Aquila, L'Aquila, Abruzzo, Italy

## ABSTRACT

**Aims:** Rice is a staple food for many countries, being fundamental for a large part of the worlds' population. In sub-Saharan Africa, its importance is currently high and is likely to become even more relevant, considering that the number of people and the per-capita consumption are both predicted to increase. The flea beetles belonging to the *Chaetocnema pulla* species group (*pulla* group), a harmful rice pest, are an important vector of the Rice Yellow Mottle Virus, a disease which leads even to 80–100% yield losses in rice production. We present a continental-scale study aiming at: (1) locating current and future suitable territories for both *pulla* group and rice; (2) identifying areas where rice cultivations may occur without suffering the presence of *pulla* group using an Ecological Niche Modelling (ENM) approach; (3) estimating current and future connectivity among *pulla* group populations and areas predicted to host rice cultivations, based on the most recent land-use estimates for future agricultural trends; (4) proposing a new connectivity index called "Pest Aggression Index" (PAI) to measure the agricultural susceptibility to the potential future invasions of pests and disease; (5) quantifying losses in terms of production when rice cultivations co-occur with the *pulla* group and identifying the SSA countries which, in the future inferred scenarios, will potentially suffer the greatest losses.

**Location:** Sub-Saharan Africa.

**Methods:** Since the ongoing climate and land-use changes affect species' distributions, we first assess the impact of these changes through a spatially-jackknifed Maxent-based Ecological Niche Modelling in GIS environment, for both the *pulla* group and rice, in two climatic/socioeconomic future scenarios (SSP_2.45 and 3.70). We then assess the connectivity potential of the *pulla* group populations towards rice cultivations, for both current and future predictions, through a circuit theory-based approach (Circuitscape implemented in Julia language). We finally measure the rice production and GPD loss per country through the spatial index named "Pest Aggression Index", based on the inferred connectivity magnitude.

**Results:** The most considerable losses in rice production are observed for Liberia, Sierra Leone and Madagascar in all future scenarios (2030, 2050, 2070). The future economic cost, calculated as USD lost from rice losses/country's GDP results are high

Corresponding author
Walter De Simone,
walter.desimone@graduate.univaq.it

for Central African Republic (−0.6% in SSP_2.45 and −3.0% in SSP_3.70) and Guinea–Bissau (−0.4% in SSP_2.45 and −0.68% in SSP_3.70), with relevant losses also obtained for other countries.

**Main conclusions:** Since our results are spatially explicit and focused on each country, we encourage careful land-use planning. Our findings could support best practices to avoid the future settlement of new cultivations in territories where rice would be attacked by *pulla* group and the virus, bringing economic and biodiversity losses.

## INTRODUCTION

Rice is one of the most cultivated crops globally, representing a fundamental source of livelihood for a large part of the world's population (*Chauhan, Jabran & Mahajan, 2017*). Its cultivation is very different in both techniques and environments, ranging from single-crop systems (temperate and tropical regions, in rainfed and irrigated conditions) to intensive monoculture (irrigated areas, in the tropics) (*Chauhan, Jabran & Mahajan, 2017*).

The African continent is very far from food self-sufficiency, a condition predicted to worsen in the next future, considering the current sociodemographic pressure (*Mbow et al., 2019*). As rice is today considered a staple food for many African countries, considering both population growth and per-capita consumption, rice crops are predicted to expand (*Mbow et al., 2019*).

Sub-Saharan Africa (SSA) comprises some of the poorest regions in the world; policy makers, donors and other partners have implemented several food strategies over the years in this area, including strengthening the rice sector, which has been identified as a vital component of food security and poverty reduction (*Jayne & Rashid, 2013*; *deGraft-Johnson et al., 2014*). Rice is rapidly gaining importance as a staple food and is now one of the major sources of food energy in SSA, being used by over 750 million people in this region (*FAO, 2017*).

The coupled effect of climate change and land-use alterations (*e.g.,* urban settlements, logging forests) is leading to biodiversity loss on both global and local scales (*Iannella, Liberatore & Biondi, 2016*; *Griscom et al., 2018*). These biodiversity variations exert pressure on agro-ecosystems, altering their functionality with consequent loss of ecosystem services, which are a part of the conditions and processes by which ecosystems (both natural and semi-natural) support and provide services to humankind. They can be divided into several categories: regulating, cultural, support, and provisioning services (*e.g.,* food, fiber, and biofuels) (*Carpenter et al., 2006*).

Agricultural production (rice, in this case) is the most clearly observable supply service, based on crop yield. Thus, the potential of rice supply as an ecosystem service can be defined as the maximum (hypothetical, considering the environmental constraints) yield of rice (*Burkhard et al., 2014*).

Rice production and the provision of related ecosystem services are highly dependent on environmental and socio-economic factors; still, some knowledge gaps occur among decision-makers for developing sustainable agriculture strategies and improving food security (*FAO, 2017*). These led to increased environmental risks (*e.g.*, drought, erosion, pollution, and pests), as well as threats of famine in developing countries (*Mbow et al., 2019*). Moreover, current climate change is affecting (and is predicted to affect) the frequency of precipitation, the UV-B radiation and intensity of droughts, directly influencing the relationships between rice and its pests and diseases (*Haq et al., 2010*). Among these last, the Rice Yellow Mottle Virus (RYMV) is the leading viral rice disease and the primary viral constraint to rice production in Africa (*Koudamiloro et al., 2015*), and different approaches were used to infer its potential spread (*Trovão et al., 2015*; *Dellicour et al., 2018*). Research has shown that, among RYMV vectors (*Traore et al., 2009*), leaf-feeders insects found in rice paddies are the main source of virus transmission (*Heinrichs, 2004*; *Koudamiloro et al., 2015*), with many studies reporting their involvement in the spread of this disease also on a landscape scale (*Heinrichs, 2004*; *Rakotomalala et al., 2019*).

Among pests, *Chaetocnema pulla* species group *sensu Biondi & D'Alessandro (2008)* (hereafter, *pulla* group), flea beetles (Chrysomelidae, Galerucinae, Alticini) occurring in most of the Afrotropical region including Madagascar, and in the Arabian Peninsula (Saudi Arabia and Yemen), are a particularly harmful pest of rice (*Oryza sativa*) (*Biondi, 2001*). Also, they represent an important vector of the RYMV in SSA (*Bakker, 1971*; *Koudamiloro et al., 2015*); moreover, *Banwo et al. (2001)* report that if specimens of *pulla* group abundantly occur when an RYMV source is present, a vast spread of the virus is expected. Rice yellow mottle virus and can cause up to 80–100% yield loss in some rice-cropping systems (*Wopereis et al., 2013*; *Asante et al., 2020*), with a decrease in the number of spikelets, the partial or total sterility of the rice panicles, and the death of the infected plant (*Kouassi et al., 2005*).

Despite the high economic importance of rice crops and its food security implication, no large-scale studies has been done to date; indeed, some local-scale research about the co-occurrence of RYMV and insects in the context of rice production was published (*Kouassi et al., 2005*; *Balasubramanian et al., 2007*; *Hubert, Lyimo & Luzi-Kihupi, 2017*). Moreover, there are no predictions of future distribution trends for pests, disease, and available rice areas, especially under climate and land-use change scenarios.

Considering these gaps in knowledge, we present a continental-scale study aiming at: (1) locating current and future suitable territories for both *pulla* group and rice; (2) identifying areas where rice cultivations may occur without suffering the presence of *pulla* group using an Ecological Niche Modelling (ENM) approach; (3) estimating current, and future connectivity among *pulla* group populations and areas predicted to host rice cultivations, based on the most recent land-use estimates for future agricultural trends;

(4) proposing a new connectivity index called "Pest Aggression Index" (PAI) that measuring the agricultural susceptibility to the potential future invasions of pests and disease; (5) quantifying losses in terms of production when rice cultivations co-occur with the *pulla* group, and identify the SSA countries which, in the future inferred scenarios, will potentially suffer the greatest losses.

## MATERIALS & METHODS

### Target species and study area

*Chaetocnema* Stephens, 1831 is a flea beetle genus widespread in all zoogeographical regions. It consists of over 300 described species. About the Afrotropical region, *Chaetocnema* genus includes over 100 known species (*Biondi, 2002*; *Biondi & D'Alessandro, 2006a*, *2006b*, *2018*). The species of this genus are often associated with moist environments and plants in many botanical families, particularly the Chenopodiaceae, Polygonaceae, Cyperaceae, Gramineae (*Biondi, Urbani & D'Alessandro, 2015*). A small group of very similar species (*Biondi & D'Alessandro, 2008*), including *Chaetocnema pulla* Chapuis, 1879, *C. nkolentangana* Bechyné, 1955, *C. subquadrata* Jacoby, 1897, and *C. vanschuybroecki Biondi & D'Alessandro, 2008*, widespread in the Afrotropical region, are here considered. The occurrence localities of *pulla* group were mainly obtained from *Biondi & D'Alessandro (2008)* and integrated with new data (M. Biondi, 2021, personal data), obtaining the most complete dataset for this taxon to date.

About the target plant species, two main species of rice are cultivated in Africa. The first is *Oryza sativa* L., native to Asia and was introduced in Africa about 450 years ago. Another lesser-known species, *O. glaberrima* Steud, is native to Africa and was domesticated in the Niger River Delta (in northern Mali) over 3,000 years ago. As a result of their evolution and domestication, both species have distinct and (*Carpenter, 1978*) complementary advantages and disadvantages for African farming systems. Asian rice (*O. sativa*) is characterized by good yields, no entrapment and crushing of grain, and high fertilizer yields, unlike its African counterpart (*O. glaberrima*). However, unlike Asian rice types, native varieties of *O. glaberrima* often have good competitiveness and weed resilience against major African biotic and abiotic stresses (*Koffi, 1980*; *Jones, Mande & Aluko, 1997*). The study area is sub-Saharan Africa, where the two target taxa both occur.

### Ecological niche modelling

To estimate the current and future climatic suitable area of both *pulla* group and rice, Ecological Niche Models were built; these have recently gained popularity among many fields of natural sciences, as they permit to better address conservation policies (*Iannella et al., 2018*; *Brunetti et al., 2019*), understand biogeographic patterns (*Iannella, Cerasoli & Biondi, 2017*; *Console et al., 2020*) and predict possible distribution of invasive species and pests (*Fick & Hijmans, 2017*).

For this aim, 19 bioclimatic variables for current climatic conditions were downloaded from the Worldclim.org repository, ver 2.1 (*Fick & Hijmans, 2017*), at 2.5 arc-min resolution. To avoid possible correlation among predictors, which may lower model's performance, both Variance Inflation Factor (VIF, threshold set = 10 following

*Guisan, Thuiller & Zimmermann (2017)*) and Pearson's r (|r| < 0.85, following *Dormann (2007)*, *Elith et al. (2006)*) were assessed through the 'vifstep' and 'vifcor' functions of the 'usdm' R package (*Naimi, 2015*), and a subset of predictors was then used for models' calibration.

Ecological Niche Models were built through the Maxent (*Phillips, Anderson & Schapire, 2006*) implementation in "SDMtoolbox" 2.4 version (*Brown, Bennett & French, 2017*), in ArcMap 10.0 (Esri, Redlands, CA, USA). This software permits to take advantage of the powerful machine-learning algorithm of Maxent and to integrate into the modelling process some details (not available on the "standard" version of Maxent), which increase models' discrimination power by lowering spatial biases. In particular, the 'Spatial jackknifing' procedure implemented in this software permits to reduce spatial autocorrelation among data by splitting the study area (and the corresponding presence occurrences and background points) into training and test datasets, iteratively (*Brown, Bennett & French, 2017*). For the present study, five spatial jackknife groups were set; moreover, considering the great extent of the study area, a bias mask to account for and correct the latitudinal bias was generated through the 'Bias File for Coordinate Data (BFCD) in MaxEnt' algorithm (*Brown, Bennett & French, 2017*) and applied to the modelling process. Before being used as input data for the models, both *pulla* group and rice occurrence data were processed through the 'Spatially Rarefy Occurence Data for SDMs' algorithm (resolution = 10 km) so as to make comparable the resolution of occurrences with one of the predictors (*Sillero & Barbosa, 2020*). To obtain reliable estimates, 10 replicates for each model parameter class were calculated, and a Regularization Multiplier = 5 was set, for a total of 250 single models for each species. Also, spatially segregated groups were chosen, as this option permits the models' training and evaluation in potentially non-analogous environments (*Brown, Bennett & French, 2017*).

Models trained on current climate were projected to future conditions using three different general circulation models (GCMs) to reduce uncertainties due to their differences (*Stralberg et al., 2015*). For this aim, 2030, 2050 and 2070 GCM projections of BCC-CSM2-MR (*Wu et al., 2019*), the IPSL-CM6A-LR (*Boucher et al., 2020*) and the MIROC6 (*Tatebe et al., 2019*) were chosen, for two Shared Socioeconomic Pathways (SSP), the SSP245 and the SSP370. These were chosen based on their link between the possible future trends of land-use (see below) and two Representative Concentration Pathways (RCPs) scenarios, with a more optimistic (RCP 4.5) and a "middle of the road" one (RCP 7.0). Moreover, the Multivariate Environmental Surface Similarity (MESS, following *Elith & Leathwick, 2009*), which measures the projected models' degree of extrapolation (*i.e.*, the divergence of environmental predictors used for models calibration with respect to the ones used for models projections), was taken into account. This information was incorporated into the MEDI, an index used to proportionally weigh future models' projections based on their degree of extrapolation measured through the MESS (*Iannella, Cerasoli & Biondi, 2017*). Models' discrimination power was assessed by both the Area Under the Curve of the Receiver Characteristic Operator obtained from the Maxent output (*Phillips, Anderson & Schapire, 2006*) and the continuous Boyce index (ranging from −1 = counter prediction to +1 = optimal prediction), which is the best

choice for evaluating presence-only ENMs (*Boyce et al., 2002*; *Hirzel et al., 2006*). This was calculated through the 'ecospat.boyce' function of the 'ecospat' package (*Di Cola et al., 2017*) in R environment (*R Core Team, 2016*).

## Land-use simulations and GIS analyses

The ENMs obtained were further analysed in a GIS environment to refine the predictions in more realistic distribution scenarios of both target taxa (*Iannella et al., 2019*). All analyses of this section were performed in QGIS 3.10.11 (*QGIS.org, 2021*).

We performed future land-use-land-cover (LULC) simulations in sub-Saharan Africa using the InVEST suite, a geospatial modelling framework tool that evaluates the impact of land-use change on ES (*Sharp et al., 2018*; *De Simone et al., 2020*).

Furthermore, we evaluated rice production under the influence of *pulla* group in different future scenarios (SSP2 and SSP3 narratives) within a time range of 40 years (2030–2050–2070).

The basic LULC we have chosen for all the analyses performed is the Global Land Cover 2010 (used as a current scenario), distributed by the Copernicus Climate Change Service (C3S), which classifies the earth's surface into 22 classes with a spatial resolution of 0.002778° (~300 m).

In addition to the map described above, to identify the rice crops more precisely, we used the SPAM (Spatial Production Allocation Model) map 2010 (*Yu et al., 2020*), isolating the "rice" class from other crops. Using GIS techniques, it was fused with C3S's "rainfed/irrigated cropland" class to create an additional land-use category (called Rice) representing potential rice crops of the current situation.

To obtain simulations of future LULC maps, the 'Scenario Generator proximity-based' (SGpb) of the InVest suite (*Sharp et al., 2018*) was used. This tool allows to create simulations of possible alternative futures using the data provided by the user.

Following the creation of the new Rice class, we translated management policies into spatial data using "guidelines" ("storylines"). To this end, the SSP was used. In particular, the SSP database (SSPdb) of *Riahi et al. (2017)* was used, which quantifies the mitigation policies. This database was created to model and compare possible future trajectories of different LULC and climatic scenarios (deriving from RCP + SSP) and quantify such changes.

Future LULC scenarios have been built to strengthen the link between the SSP narratives and the new RCP scenarios, to have more precise information about future projections.

The socio-economic scenarios chosen were SSP2 4.5 and SSP3 7.0, and the new Rice class previously created was selected for the future simulations. Given the unavailability of the SSP3 7.0 scenario in SSPdb, we used the baseline SSP3 data because *Fujimori et al. (2017)* indicated that the case with a forcing level of 7.0 W/m$^2$ roughly corresponds to the SSP3 baseline cases.

To extract the real values of the rice crops area for our study region, we used the FAOSTAT database (*FAO, 2016*). To obtain the future growth rate of rice crops area, we

performed a linear regression analysis on the data obtained by FAOSTAT (1973 ÷ 2010). We then performed a ratio between the cultivated areas of the SSPdb and those of the current LULC to obtain the data for future simulations of the land-use scenarios.

We considered the Croplands variation in millions of hectares for two SSP narratives: SSP2 and SSP3. In SSP2 (midway), land-use change is incompletely regulated, *i.e.*, tropical deforestation continues, albeit with slow decline rates; crop yield rates slowly increase over time (*Fricko et al., 2017*).

The second SSP narrative considered, SSP3 (regional rivalry), shows a resurgent nationalism, concerns about competitiveness and security, and pushes countries to focus more and more on national or regional issues. Land-use change is hardly regulated and crop yield rates decline sharply over time, mainly due to the minimal transfer of new agricultural technologies to developing countries (*Fujimori et al., 2017*).

The values for the correct setting of the SGpb tool were obtained by calculating the variation (2010 ÷ 2030; 2030 ÷ 2050; 2050 ÷ 2070) between the present and future rice areas.

To simulate the future LULC, we applied the two SSP projections to cultivated land in the LULC classes that are most likely to be modified (see below) closer to existing cultivated land. Three types of land cover were chosen: (a) "Focal ground cover" (SGpb tool converts classes from the edge of focal land cover zones) was chosen based on the most likely categories to increase agricultural areas. It is more likely to extend agriculture from existing crops than to create new ones. (b) "Convertible land cover" represents the types of land cover that can be converted, chosen based on the classes that have changed the most in the last 20 years, and obtained from the analysis of these LULC variations (*ESA, 2017*). (c) "Replacement land cover" was chosen because the agricultural land cover class was the most frequently converted compared to the other LULC categories (*De Simone et al., 2020*).

To identify the areas of co-occurrence for the correct MEDI models (*pulla* group and rice), we polygonized and intersected them with the simulations of future scenarios (LULC_2030_SSP2_4.5, LULC_2050_SSP2_4.5, LULC_2070_SSP2_4.5, LULC_2030_SSP3_7.0, LULC_2050_SSP3_7.0 and LULC_2070_SSP3_7.0).

From the spatial data obtained, we calculated the resulting rice crops area values to estimate the surface area most affected by the pest and to evaluate the co-occurrence between the two taxa within each sub-Saharan African country.

## Landscape connectivity

To assess the potentiality of the study area in allowing the current and future colonization of rice cultivations from where *pulla* group currently occurs (*i.e.*, measuring landscape connectivity for pest invasion), Circuitscape v.5 (*Anantharaman et al., 2020*) in Julia 1.5.3 programming language (*Bezanson et al., 2017*) was used.

Circuitscape applies both random walks and circuit theory to model the ecological connectivity in a target area (*McRae et al., 2008*; *McRae, Shah & Edelman, 2016*) and is particularly useful and robust, with respect to other connectivity software, for applications

in which the species has no previous knowledge of the route to use for the colonization (*McClure, Hansen & Inman, 2016*), as in this case.

For this aim, the ENMs outputs were converted into friction maps (the inverse of suitability maps) through the 'Raster calculator' tool in ArcMap 10.0 and used as resistance layers (*i.e.*, the resistance which a landscape has to the movement of target individuals/taxa). The 'Advanced' mode was used to enable the activation of independent sources (*pulla* group occurrence localities) and grounds (the destination patches, in our case, the current and future predicted rice cultivations in raster format). Considering the great geographic extent of the study area, the 'solver = cholmod' (a solver method applying a Cholesky decomposition) was used (*Bezanson et al., 2017*). The outputs obtained are raster maps depicting a current value, which were subsequently managed in ArcMap 10.0.

### Pest aggression index (PAI) and provisioning ecosystem services

To go further into the understanding of the consequences of the connectivity which the landscape offers to the invasion of *pulla* group to rice cultivations, we developed an appropriate index, which we call "Pest Aggression Index" (PAI), calculated through the corridor edge (CE) and Mean Corridor Magnitude (MCM, whose rationale is reported in Fig. 1), the parameters of which are introduced and detailed below.

When referred to target countries ($\text{PAI}_{Country}$), to obtain a single value for a target area (in our case, a country, even though the PAI can be applied to any territory), we first process the connectivity maps inferred in the previous steps. Considering that these are continuous rasters spanning the whole study area (Fig. 1A), we remove zero values to isolate each corridor from another (Fig. 1B). Then we calculate the mean cell raster value in R (*R Core Team, 2016*), obtaining a single number that represents the magnitude of an ideal vector which symbolizes the corridor, a value we name Mean Corridor Magnitude (MCM) (Fig. 1C). The MCM is then used to set the threshold of the continuous corridor rasters. Concurrently, we consider the length (in km) of the corridor edge (CE), which is in contact with the destination rice patch, representing the "frontline" which the pests will first find, moving through the landscape corridor (Fig. 1C).

To standardize the results and compare the PAI for each country, we refer to the multiplication of these first two terms to the total rice cultivation (RP) and agricultural area (AR) of the target country through the formula:

$$\text{PAI}_{Country} = \frac{\sum_{i=1}^{n} CE_i * \text{MCM}_i}{Total\ RP_{Country}\ *\ Total\ AR_{Country}}$$

The $\text{PAI}_{Country}$ is divided by the maximum value of each *year_SSP* scenario and further normalized by subtracting the obtained value from one. Then, the value obtained is multiplied by the rice production of each country, obtaining a "weighted" one, which mirrors the potentiality that the *pulla* group populations have to reach and affect cultivations.

This index ranges from zero (no landscape connectivity among the pest's populations and target cultivations) to one (the highest PAI, the highest expected losses to cultivations in each country).

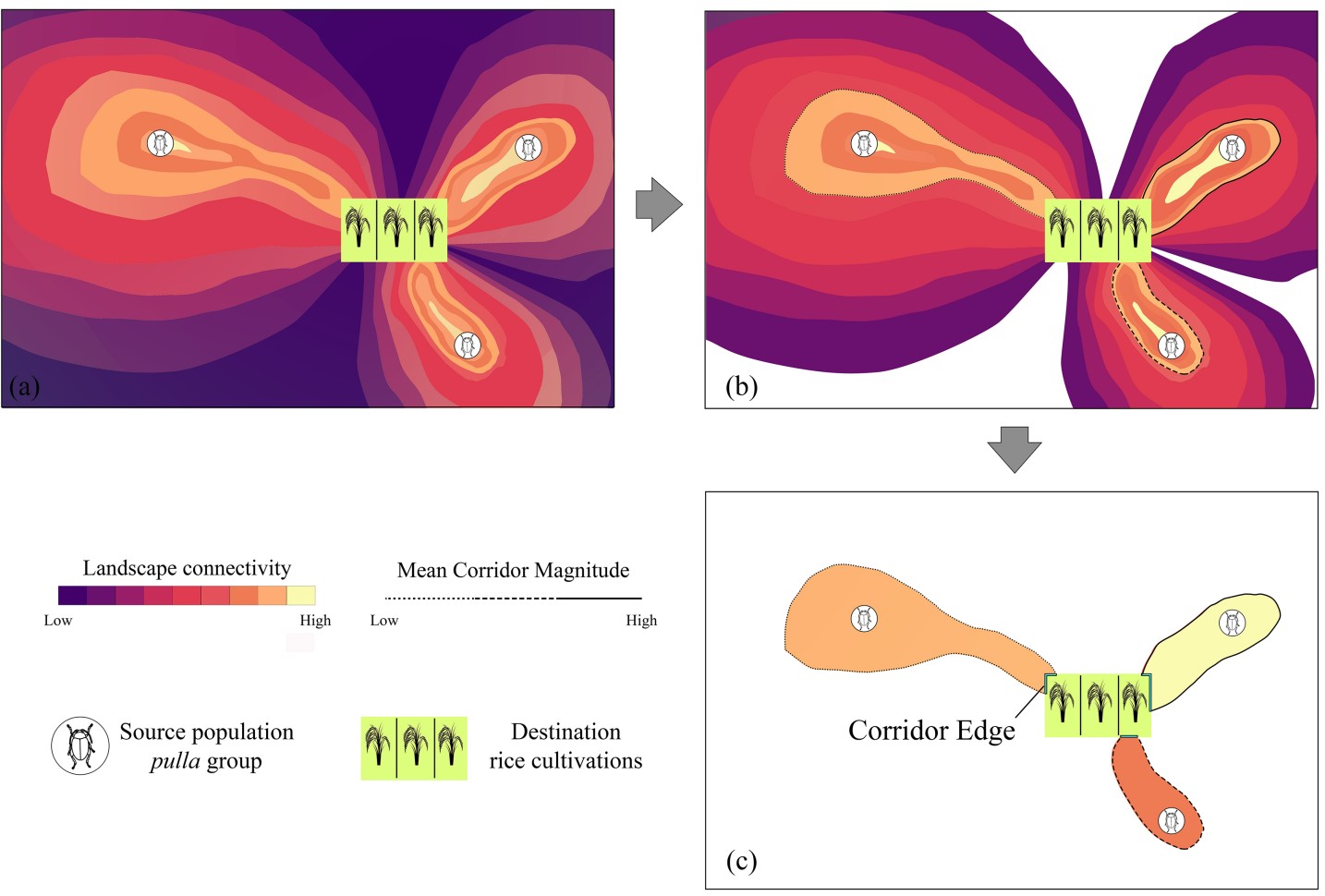

**Figure 1 Mean Corridor Magnitude workflow.** The rationale of the workflow used to calculate the Mean Corridor Magnitude, with the total landscape connectivity among *Chaetocnema pulla* populations and rice cultivations (A) being discretized in single corridors (B) and converted into unique connectivity values, from *C. pulla* populations to rice cultivations, based on the mean connectivity value such obtained (Mean Corridor Magnitude) (C).

Considering the data achieved from the previous analyses, we calculated the future potential rice production (FPRP) in the different scenarios (SSP2 4.5 2030–2050–2070 and SSP3 7.0 2030–2050–2070) for each sub-Saharan country. We calculated the FPRP by multiplying the areas of the rice patches (in co-occurrence with the *pulla* group) with the average rice yield (ton/ha). As previously done to obtain the growth rate of rice crops area, to calculate the rice yield's future growth rate, we performed a linear regression analysis on the data obtained by FAOSTAT (1973 ÷ 2010).

To obtain the final values of loss in rice production (provisioning ecosystem service), we calculated for each country the difference between the total FPRP (without *pulla*) and the FPRP values obtained under the influence of *pulla*. Furthermore, to standardize these values, we multiplied the final value by the PAI values (by country and LULC scenario). We have condensed the loss in rice production (%) obtained by the countries in a heatmap achieved through the Seaborn library in the Python environment (*Waskom, 2021*).

To achieve a more accurate view of the economic losses by country, we also calculated the ratio between the economic loss and the gross domestic product (GDP) for each nation. We used the obtained economic loss data and the projected GDP values to the future taken from the SSPdb, Faostat and WFP VAM (*World Food Programme (WFP), 2020*) database. A selection of countries was made both for the results obtained and for the availability of the various data sources (rice price, GDP).

## RESULTS

### Models calibration and evaluation

After the variables' selection process, nine bioclimatic variables were selected for the models' calibration, namely BIO1 (mean annual temperature), BIO2 (mean diurnal range), BIO3 (isothermality), BIO8 (mean temperature of wettest quarter), BIO13 (precipitation of the wettest month), BIO14 (precipitation of driest month), BIO15 (precipitation seasonality), BIO18 (precipitation of warmest quarter) and BIO19 (precipitation of coldest quarter). The models built on this set of variables resulted in high discrimination power, with an AUC = 0.817 for *Oryza* and AUC = 0.807 for *pulla* group (Fig. S1A) and the Continuous Boyce Index obtained for *Oryza* = 0.983 and for *pulla* group = 0.957 (Fig. S1B). Moreover, low spatial standard deviation resulted for both the models calibrated on the current climatic conditions (Figs. S1C, S1D).

### Current distribution and future co-occurrences

The occurrence localities of *pulla* group are clustered in specific areas, mainly Central-Eastern and Western Africa, as well as Madagascar, while rice is widely cultivated (obtained from *GBIF (2020)*; accessed on November 18, 2020), even in territories where *pulla* group does not occur (Fig. 2).

These distributions are partially mirrored in the areas obtained through the ENMs. The combinations of the low, medium, and high suitability classes report broad areas in which cultivations in both Western and some territories of Eastern Africa may not currently suffer the presence of *pulla* group (Fig. 3).

The general trend of vast areas hosting the conditions mentioned above disappears in future projections (Figs. 4A–4F). Instead, an increase in the Mid-Mid, Mid-High and High-Mid suitability classes (intended as *pulla*-rice) is observed. For instance, the SSP_2.45 shows a +68% of the Mid-High class in the 2030 scenario (Fig. 4A), with some peaks observed in the 2070 SSP_3.70, where the Mid-Mid class reaches an increase of +84% (Fig. 4F).

### Connectivity of pests to rice cultivations and provisioning ecosystem services

The landscape connectivity inferred from future scenarios show corridors allowing *pulla* group to reach rice cultivations in specific African territories, such as Nigeria, Cameroon, Sudan, South Sudan, Democratic Republic of Congo, Zimbabwe, Republic of South Africa, and Madagascar (Fig. A.2).

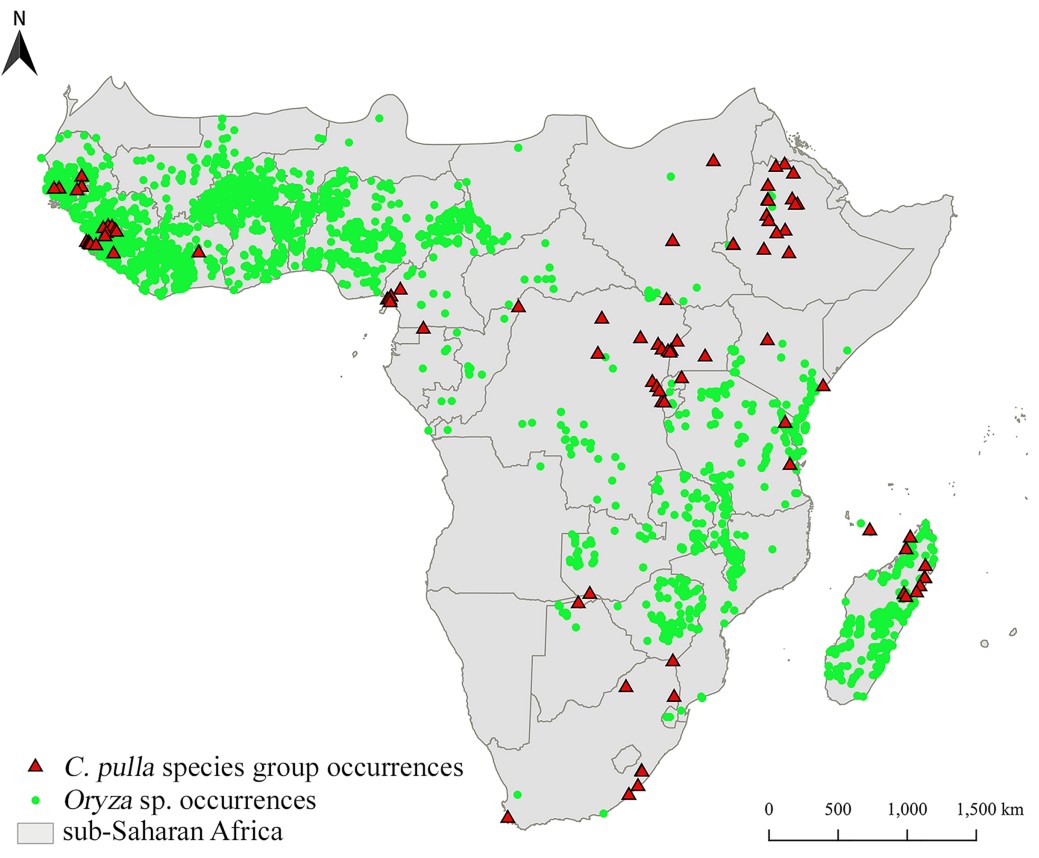

**Figure 2 Distribution of two target taxa.** Distribution of *Chaetocnema pulla* species group and rice cultivations in sub-Saharan Africa.

The Pest Aggression Index (PAI) values ranged from 0 to 0.992; these values were used to obtain the rice production loss (in percent) for each SSA country, as summarized in the heatmaps reported in Figs. 5A, 5B. Different results are observed according to the SSP scenario (2.45 and 3.70). The production loss for the SSP 2.45 and SSP 3.70 narratives (Figs. 5A, 5B) shows that the countries that will suffer the largest production losses (>−90%) in all the scenarios analyzed (2030, 2050, 2070) are Liberia and Sierra Leone. In the SSP 2.45 scenario, other countries such as Madagascar, Guinea, Côte d'Ivoire, Ghana and Guinea–Bissau (only in 2070) show losses between 65% to 83%. In the SSP 3.70 scenario (Fig. 5B), the same countries recorded a similar loss of rice production to the SSP 2.45 scenario, with the addition of the Central African Republic, Togo and Guinea–Bissau. Some countries, such as the Democratic Republic of Congo and Cameroon, show average loss values (46–54%) only in the SSP 3.70_2070 scenario.

Concerning the ratio between production loss and GDP (Figs. 5A, 5B; stacked-bars), the condition is still different, as emerging from the heatmap. In the SSP 2.45 narrative (Fig. 5A), the countries showing the most considerable losses (~−0.95%) are the Central African Republic, Guinea–Bissau, Liberia, Madagascar, Gabon, the Democratic Republic of Congo and Mozambique (respectively, in descending order). In the SSP 3.70 scenarios

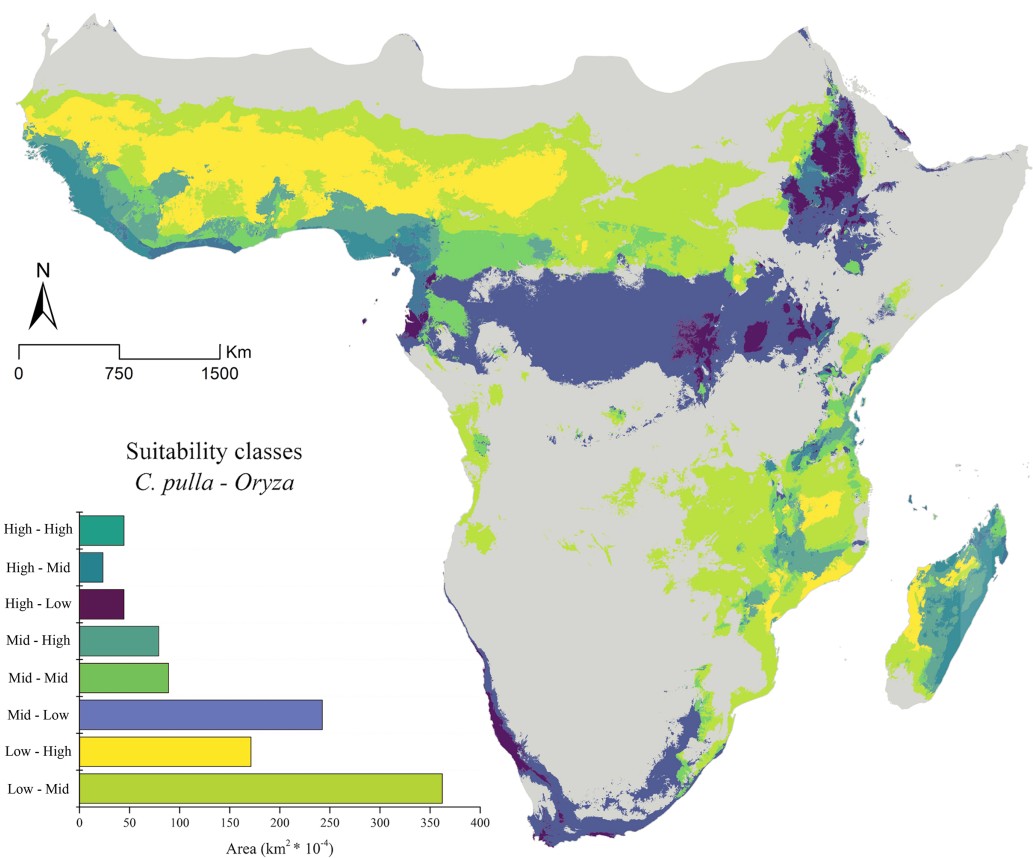

**Figure 3 Current suitability classes' combinations.** The current asset of suitability classes' pairs between *Chaetocnema pulla* species group and rice cultivations.

(Fig. 5B), harsh economic losses are predicted to reach a net value of −4.8%, with the Central African Republic (highest absolute values), Guinea–Bissau, the Democratic Republic of the Congo, Chad, Liberia, Mozambique, Madagascar and Gabon being the more affected by the loss. The Central African Republic is the only country showing a notable increase in 2070 (+0.82%) (Fig. 5B).

## DISCUSSION

Our results show a diversified, complex geographic asset of rice's suitable territories with respect to the ones of the *pulla* group, its virus-bearing pest. Rice is cultivated in different environments, from hot and humid alluvial plains to high mountain ranges (up to about 2,700 m), from the equator to high latitudes (about 53 degrees), in four different farming systems (rain-fed lowland, lowland, irrigated, upland rice and mangrove-rice fields) (*Oteng, Sant'Anna & Van Tran, 1999*). In SSA, about 40% of rice cultivations occur in the uplands (contributing 19% to total rice production, thus attaining the lowest potential yields), 37% in the rainfed lowlands (contributing 48% to total rice production) and 14% in the irrigated territories (contributing 33% to total rice production); the remaining 9% is managed through the mangrove-rice system (*Seck et al., 2010*).

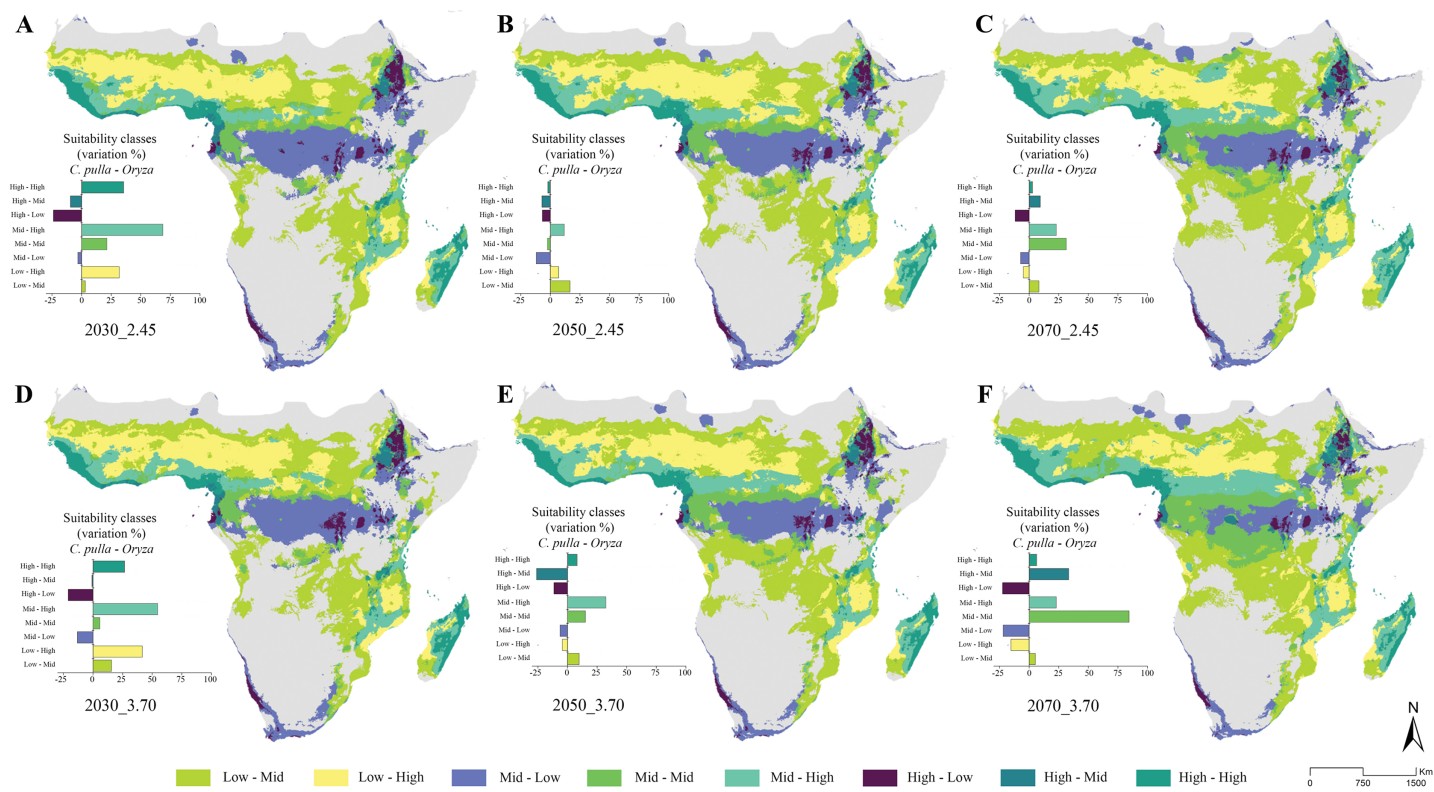

**Figure 4 Future suitability classes' combinations.** (A–D) Future predicted assets of suitability classes' pairs between *Chaetocnema pulla* species group and rice cultivations for all considered scenarios.

SSA rice production is generally profitable and competitive in irrigated cultivations compared to imported rice. Considering these encouraging results, SSA countries increased the investment in constructing new irrigation systems (*Seck et al., 2010*). Irrigation of rice cultivations allowed farmers to obtain very high yields (3–4 times higher than rainfed rice) and increased the harvests' intensity (*Seck et al., 2010*). Unfortunately, despite having higher yields, these crop systems tend to be more susceptible to several pests and diseases, such as the rice yellow mottle virus, with high temperatures (which are predicted to increase worldwide by the IPCC) positively influencing most of them (*Haq et al., 2010*).

In this context, we report some key results in which both climate and LULC current and future scenarios heavily influence rice and *pulla* group co-occurrence. Indeed, our current predictions show that the climatic suitability in the current scenario for rice crops is mainly found in two districts, Western and Eastern Africa, being in accordance with *Arouna et al. (2021)*. The former shows high suitability for the *pulla* group, while the latter is suitable to this pest only in its northern part, posing at-risk rice crops in both cases. Madagascar rice cultivations are also highly susceptible to *pulla* group attacks. This susceptibility condition is predicted to worsen in the future. Despite peaking in 2030 with

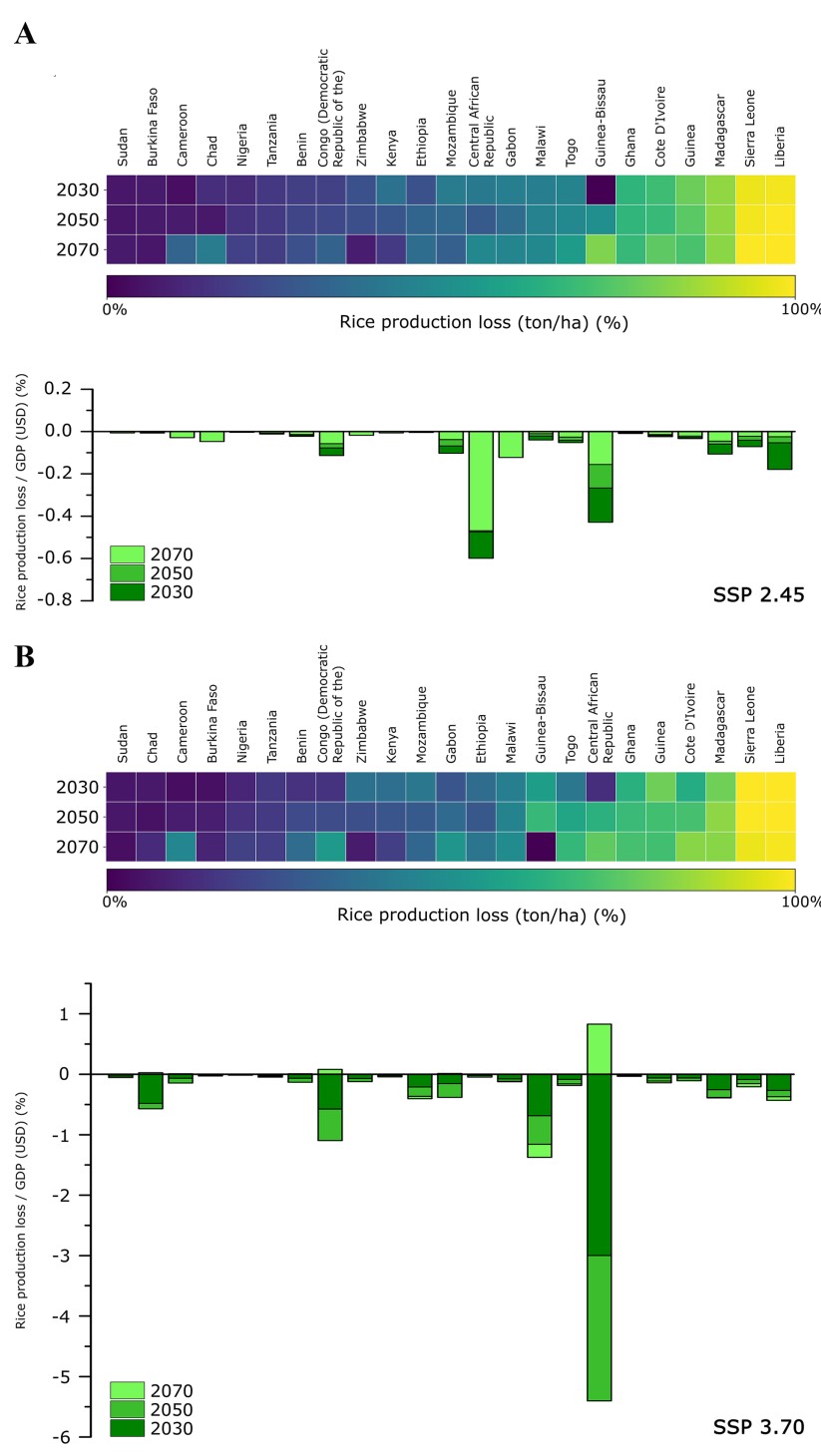

**Figure 5 Losses in rice production and gross domestic product (GDP).** Heatmap of rice production loss (ton/ha * year) expressed as a percentage for each sub-Saharan country (for all future scenarios); stacked-bars of rice production loss in relation to GDP in USD (expressed in %) for each sub-Saharan country (all future scenarios considered). (A) SSP_2.45 scenario; (B) SSP_3.70 scenario.

the Mid-High (*pulla*-rice) class, the increase of the crucial suitability classes (Mid-Mid, Mid-High, High-Mid, High-High) appears to stabilize within a range between +25% and +30% in the SSP 2.45. Conversely, in the SSP 3.70, the suitable area of these suitability classes is predicted to increase from 2030 until 2070, sometimes with high peaks (Mid-Mid class, 2070).

These future highly suitable co-occurrences are also consistent with the future high connectivity found among *pulla* group populations and rice patches, which is predicted to allow the pest (bearing the RYMV) to infest the rice cultivations. Even though this virus can be transmitted through contact by many mechanisms (*Abo et al., 2000*; *Sarra et al., 2004*; *Traore et al., 2009*), the presence of beetles is assumed to support long-distance transmission (*Fargette et al., 2006*). As further evidence, *Banwo et al. (2001)* found that even only four individuals of *pulla* group determine the 80% of probability of RYMV infection, with a corresponding high yield loss (*Asante et al., 2020*).

This is reflected by the rice production loss analyses (heatmaps), in which we find some countries to suffer high losses, with the greatest predicted for Liberia and Sierra Leone. Despite this data, these countries are reported to suffer less economic damage due to their cropping systems (uplands rice crops), which are less susceptible to *pulla* group attack (*Mogga et al., 2012*; *Oludare et al., 2016*; *Suvi, Shimelis & Laing, 2019*). In fact, according to *Oludare et al. (2016)*, uplands crops can resist RYMV attack better than lowland and irrigated croplands, which are instead widely used in Central Africa Republic, Madagascar, and Guinea. In fact, the economic analysis highlights critical issues regarding these countries in all future scenarios considered. A few exceptions where a positive trend in the rice production loss/GDP ratio is predicted (Central African Republic and Democratic Republic of Congo) are probably due to the lesser influence of *pulla* group on the crops of these two countries. This is a result of the lower PAI values found, which are in turn caused by a local decrease of suitability due to climate and land-use change.

## CONCLUSION

The ongoing climate and land-use changes are proven to be detrimental to many processes, especially in terms of human-nature interactions. Among these, agricultural practices standing their foundations on the equilibria of plant-pest relationships are predicted to be particularly threatened in the future, with severe economic losses, as reported in our research.

The use of combined ecological modelling, GIS analyses and connectivity estimates, summarized in the Pest Aggression Index and presented in this work, is applicable in any plant/pest context, even where few spatial data (*e.g.*, occurrence-only and environmental predictors' datasets) is available.

Considering the explicit spatial reference, this approach could target specific actions and future landscape management, such as careful land-use planning to maintain biodiversity and focus on economic investments.

### Funding

This work was supported by the A.I.M. Project—PON R & I 2014–2020 No. 1870582. The funders had no role in study design, data collection and analysis, decision to publish, or preparation of the manuscript.

### Grant Disclosures

The following grant information was disclosed by the authors:
A.I.M. Project—PON R & I 2014–2020: 1870582.

### Competing Interests

The authors declare that they have no competing interests.

### Author Contributions

- Mattia Iannella conceived and designed the experiments, performed the experiments, analyzed the data, prepared figures and/or tables, authored or reviewed drafts of the paper, and approved the final draft.
- Walter De Simone performed the experiments, analyzed the data, prepared figures and/or tables, authored or reviewed drafts of the paper, and approved the final draft.
- Paola D'Alessandro analyzed the data, authored or reviewed drafts of the paper, and approved the final draft.
- Maurizio Biondi conceived and designed the experiments, authored or reviewed drafts of the paper, and approved the final draft.

### Data Availability

All the data used in this work are available in Supplementary File.

### Supplemental Information

Supplemental information for this article can be found online at http://dx.doi.org/10.7717/peerj.12387#supplemental-information.

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
