# Peer review of "Climate change favours connectivity between virus-bearing pest and rice cultivations in sub-Saharan Africa, depressing local economies"

_PeerJ, doi:10.7717/peerj.12387_

## Round 0.1 · original submission · Major Revisions

Your manuscript has been reviewed by experts in the field and we request that you make major revisions before it is processed further.

Reviewer 1 ·

Basic reporting

The paper is very interesting but the presentation requires minor revision because of the following.
1. Semi-colon is overused. In most of the use cases, the authors can just replace them with a period (.) and create two simple sentences instead. As a result, there are many long sentences separated by semi-colons (;). This makes it hard to read.
2. There are several editorial revisions I have highlighted (with suggestions) in the annotated PDF.

Experimental design

The paper is great in its experimental design. However, I would like to seek some clarifications from the authors.

Preamble: The fusion of the 2010 C3S land cover map and the SPAM’s 2016 map is acceptable only when data is unavailable. If you refer to this link (http://2016africalandcover20m.esrin.esa.int/) you will find the 2016 land cover map at 20-m spatial resolution produce from Sentinel-2 by the European Space Agency.

Question: What was the reason for fusing the 2010 C3S land cover map with the SPAM’s 2016 map? Which year then became the baseline for the studies, 2010 or 2016?

Validity of the findings

no comment

Additional comments

General comment: This paper is very interesting and has policy relevance in SSA in respect of food security. Congratulations to the authors for the great job done.

Suggestions for improvement:
1. The authors should justify in the paper why they fused data from the years 2010 and 2016 and explain the reason, for example, to choose 2010 as a baseline instead of 2016?
2. The interpretation of Figure 5b appears slightly different from what the figure shows (L366). Authors are to check and make changes if required.
3. Please provide further information about the nineteen bioclimatic variables so that others verify results in future studies. What exactly are the variables?
4. Refer to the annotated PDF for more specific revision suggestions.

Annotated reviews are not available for download in order to protect the identity of reviewers who chose to remain anonymous.

Reviewer 2 ·

Basic reporting

The authors are specialists of the biology of the flea beetles Chaetocnema pulla species group (pulla group), a harmful rice pest vector of rice yellow mottle virus, an important pathogen of rice in Africa. In this study, they implement a large range of statistical techniques, not necessarily familiar to the biologists, to assess (i) the impact of ongoing climate and land-use changes for both the pulla group and rice in two climatic/socioeconomic future scenarios through ecological niche modelling, (ii) the connectivity potential of the pulla group populations towards rice cultivations for both current and future predictions, through a circuit theory-based approach, (iii) the rice production and GPD loss per country through the spatial index named “Pest Aggression Index”, based on the inferred connectivity magnitude.

These are quite important objectives, not attainable through direct biological experimentation. Additional information on the datasets used in this study on Chaetocnema pulla, on rice in Africa, and on the impact of the virus on rice would be useful. How the degree of precision and the biases of the different datasets impact the conclusions should be discussed. Otherwise, some conclusions (especially those of figure 5) may be questionable. These queries are detailed below.

Data used in figures 2, 3 and 4.
Chaetocnema pulla. The authors are specialists of Chaetocnema pulla. Does it mean that the insect is not present in the areas where it is not reported on figure 2? For instance, in the Niger Inner Delta (in Mali) which is the region of diversification of the virus and where damaging epidemics often occur? If so (i.e., the insect is not present), this would weaken the putative link between Chaetocnema pulla and rice yellow mottle (see remark below). If not (the insect is present but has not been reported), how this sample bias affect the conclusions?

Rice. There are several maps available on rice cultivation in Africa. That used in Dellicour et al 2018 to assess the link between rice yellow mottle spread and rice cultivation was generated from You et al 2009 (reference below). It displayed rice distribution patterns sometimes different from that of the study. A justification of the map used in this study (vs that used in Dellicour et al for instance) would be useful. Would different conclusions on the relationships between Chaetocnema pulla and rice be reached if the You et al map had been used?

Data used in figure 5.
Recent studies showed that beetles are not necessarily the main vector of rice yellow mottle (see Traoré et al 2009 for a review), and underlined the role of transmission through cultural practices. The means of transmission are quite variable between regions and years. Morevoer, several transmission means are involved simultaneously. These results are at odds with the sentence “ the presence of pulla group mostly determines the infection” (line 404). Are these observations on the complexity and the variability of the ecology of the disease taken into account in the study? How is the associated uncertainty incorporated in the analyses?

The link between rice yellow mottle infection and rice loss is highly dependent on the cultivar used, the date of infection, the climatic conditions… Are these observations on the complexity of the link between the disease and the crop loss, as well as the scarcity of information available on these relationships taken into account in the study? How is the associated uncertainty incorporated in the analyses?

Methodology. There are several sources of uncertainties and biases in the datasets used (see above). Were negative controls - such as those used in Dellicour et al 2018 – implemented to test the significance of the relationships obtained in the study? Would it be possible to calculate a a confidence interval for the quantitative conclusions?

References
The list of references on rice yellow mottle epidemiology should be enlarged.

References quoted in the review
Dellicour et al. (2018) ‘On the Importance of Negative Controls in Viral Landscape Phylogeography’, Virus Evolution, 4: 23.
HarvestedChoice (2011) Rice Area Harvested (ha) (2000). Washington, DC: International Food Policy Research Institute and St. Paul, MN: University of Minnesota. http://harvest choice.org/node/4799
Traore´, O. et al. (2009) ‘A Reassessment of the Epidemiology of Rice Yellow Mottle Virus following Recent Advances in Field and Molecular Studies’, Virus Research, 141: 258–67.
You, L., Wood, S., and Wood-Sichra, U. (2009) ‘Generating Plausible Crop Distribution Maps for Sub-Saharan Africa Using a Spatially Disaggregated Data Fusion and Optimization Approach’, Agricultural Systems, 99: 126–40.

Experimental design

not applicable

Validity of the findings

not applicable

Additional comments

not applicable

---

## Round 0.2 · accepted · Accept

I agree with the reviewer. I consider this version is ready for publication.

Reviewer 1 ·

Basic reporting

no comment

Experimental design

no comment

Validity of the findings

no comment

Additional comments

I'm happy about the changes made in the revised manuscript. Congratulations to the authors.

Reviewer 2 ·

Basic reporting

In the rebuttal letter, the authors replied satisfactorily to the queries I raised. They made the necessary modifications in the manuscript.

Experimental design

OK

Validity of the findings

OK

Additional comments

none